# New Cyclam-Based Fe(III) Complexes Coatings Targeting *Cobetia marina* Biofilms

**DOI:** 10.3390/molecules30040917

**Published:** 2025-02-16

**Authors:** Fábio M. Carvalho, Luciana C. Gomes, Rita Teixeira-Santos, Ana P. Carapeto, Filipe J. Mergulhão, Stephanie Almada, Elisabete R. Silva, Luis G. Alves

**Affiliations:** 1LEPABE—Laboratory for Process Engineering, Environment, Biotechnology and Energy, Faculty of Engineering, University of Porto, Rua Dr. Roberto Frias, 4200-465 Porto, Portugalritadtsantos@fe.up.pt (R.T.-S.);; 2ALiCE—Associate Laboratory in Chemical Engineering, Faculty of Engineering, University of Porto, Rua Dr. Roberto Frias, 4200-465 Porto, Portugal; 3BioISI—Biosystems & Integrative Sciences Institute, Faculdade de Ciências da Universidade de Lisboa, Campo Grande, 1749-016 Lisboa, Portugal; apcarapeto@fc.ul.pt (A.P.C.);; 4Departamento de Física, Faculty of Sciences, University of Lisboa, Campo Grande, 1749-016 Lisboa, Portugal; 5Centro de Química Estrutural, Instituto Superior Técnico, Universidade de Lisboa, Av. Rovisco Pais 1, 1049-001 Lisboa, Portugal; 6Departamento de Química e Bioquímica, Faculty of Sciences, University of Lisboa, Campo Grande, 1749-016 Lisboa, Portugal; 7Centro de Química Estrutural, Institute of Molecular Sciences, Associação do Instituto Superior Técnico Para a Investigação e Desenvolvimento, Av. António José de Almeida nº12, 1000-043 Lisboa, Portugal

**Keywords:** marine biofouling, cyclam-based complexes, biofilm architecture, polyurethane marine paint, antifouling coating

## Abstract

Recent research efforts to mitigate the burden of biofouling in marine environments have focused on the development of environmentally friendly coatings that can provide long-lasting protective effects. In this study, the antifouling performance of novel polyurethane (PU)-based coatings containing cyclam-based Fe(III) complexes against *Cobetia marina* biofilm formation was investigated. Biofilm assays were performed over 42 days under controlled hydrodynamic conditions that mimicked marine environments. Colony-forming units (CFU) determination and flow cytometric (FC) analysis showed that PU-coated surfaces incorporating 1 wt.% of complexes with formula [{R_2_(^4-CF3^PhCH_2_)_2_Cyclam}FeCl_2_]Cl (R = H, HOCH_2_CH_2_CH_2_) significantly reduced both culturable and total cells of *C. marina* biofilms up to 50% (R = H) and 38% (R = HOCH_2_CH_2_CH_2_) compared to PU-coated surface without complexes (control surface). The biofilm architecture was further analyzed using Optical Coherence Tomography (OCT), which showed that biofilms formed on the PU-coated surfaces containing cyclam-based Fe(III) complexes exhibited a significantly reduced thickness (58–61% reduction), biovolume (50–60% reduction), porosity (95–97% reduction), and contour coefficient (77% reduction) compared to the control surface, demonstrating a more uniform and compact structure. These findings were also supported by Confocal Laser Scanning Microscopy (CLSM) images, which showed a decrease in biofilm surface coverage on PU-coated surfaces containing cyclam-based Fe(III) complexes. Moreover, FC analysis revealed that exposure to PU-coated surfaces increases bacterial metabolic activity and induces ROS production. These results underscore the potential of these complexes to incorporate PU-coated surfaces as bioactive additives in coatings to effectively deter long-term bacterial colonization in marine environments, thereby addressing biofouling-related challenges.

## 1. Introduction

Marine biofouling remains an area of intense research due to its significant environmental and economic impacts. Biofouling on ship hulls increases frictional drag, potentially raising fuel consumption by up to 40%, and causing subsequent power penalties of up to 86% at cruising speed, depending on the size and type of the vessel [1,2]. This additional drag requires more energy for propulsion, leading to higher fuel use and increased emissions of hazardous and greenhouse gases such as CO_2_, SO_2_, and NO_x_ [3,4]. Furthermore, biofouling facilitates the transport of nonindigenous organisms across ecosystems, including potentially pathogenic species, thereby disturbing local biodiversity and sometimes causing ecological imbalances [5,6].

At the heart of biofouling issues lies the formation of biofilms, which are complex microbial communities that adhere to surfaces in aqueous environments. Biofilms are central not only to biofouling but also to Microbially Influenced Corrosion (MIC), which is estimated to account for up to 20% of corrosion in aqueous systems, causing billions of dollars in damage annually across industries [7,8]. The presence of biofilms promotes MIC by accelerating corrosion processes, thus shortening the lifespan of materials and equipment, and increasing maintenance and repair costs. In recent years, many antifouling strategies have been explored to develop efficient and environmentally friendly marine coatings to replace the current toxic biocide-based coatings [9,10]. These include the development of coatings with natural compounds as antifouling agents, self-polishing coatings, biodegradable polymer coatings, low-surface-energy coatings, amphiphilic coatings, and grafted bioactive agents in coatings’ polymeric matrix [7,8,9,10,11]. This research is crucial for protecting submerged surfaces from marine biofouling and MIC, as well as preventing their undesirable consequences.

Due to their proven effectiveness, biocide-containing paints, primarily formulated with copper or zinc as active ingredients, remain the most widely used and commercially established marine antifouling coatings [12,13,14]. Nevertheless, stricter international environmental regulations are driving the development of alternative protective coatings. Cyclam-based compounds, known for their biocompatibility and strong metal chelation properties, have been explored as antimicrobial agents for biomedical applications owing to their metal complexes with zinc, nickel, cobalt, manganese, copper, and iron [15,16]. Although microorganisms require essential metal ions for physiological functions, the excess of these ions can have detrimental effects on microbial cells, including membrane degradation, protein dysfunction, and oxidative stress [17,18,19]. Regarding iron metal complexes, they have been documented to impact bacterial cells by causing oxidative stress, inhibiting respiratory processes and ATP production, increasing cell hydrophobicity, and facilitating cell wall penetration [18,20]. One potential mechanism for bacterial death by iron(III) complexes is the overproduction of reactive oxygen species (ROS), such as hydrogen peroxide and hydroxyl radicals [18,21,22]. The oxidative stress induced by these radicals can cause severe bacterial damage, such as lipid peroxidation in cell membranes, protein and DNA damage, and ultimately, cell death [18,22]. Additionally, the generation of ROS can disrupt the efflux pumps through the disturbance of the proton gradient and membrane potential [20], leading to membrane disruption. Moreover, since iron ions are essential for bacterial growth, the uptake of iron–cyclam conjugates can inadvertently occur through the natural bacterial processes for iron acquisition, where Fe^3+^ acts as a carrier for the bioactive ligands, thereby increasing their concentration inside bacterial cells, and improving their antimicrobial effectiveness [17]. Cyclam-based Fe(III) complexes have already demonstrated potent antifungal activity against *Candida krusei* and *Cryptococcus neoforms* [16]. In this study, we investigated for the first time the formulation of novel polyurethane (PU)-based coatings incorporating cyclam-based Fe(III) complexes of formula [{H_2_(^4-CF3^PhCH_2_)_2_Cyclam}FeCl_2_]Cl (**FeCy-1**) and [{(HOCH_2_CH_2_CH_2_)_2_(^4-CF3^PhCH_2_)_2_Cyclam}FeCl_2_]Cl (**FeCy-2**), as shown in Figure 1. Additionally, we demonstrated the potential of these FeCy complexes to function as bioactive agents when integrated into polymeric coatings, effectively preventing colonization by marine microorganisms. The antifouling performance was tested against *Cobetia marina* as a model proteobacteria in marine biofouling research.

## 2. Results and Discussion

In this study, cyclam-based Fe(III) complexes of formula [{H_2_(^4-CF3^PhCH_2_)_2_Cyclam}FeCl_2_]Cl, **FeCy-1**, and [{(HOCH_2_CH_2_CH_2_)_2_(^4-CF3^PhCH_2_)_2_Cyclam}FeCl_2_]Cl, **FeCy-2**, were used as bioactive additives in polyurethane-based coating formulations. The antifouling potential of the PU-based coating surfaces with 1 wt.% of **FeCy-1** (**PU/FeCy-1**) and **FeCy-2** (**PU/FeCy-2**) was assessed against *C. marina* for 42 days under hydrodynamic conditions mimicking those commonly encountered in marine environments.

The biofilm cell viability on **PU** (control), **PU/FeCy-1,** and **PU/FeCy-2** surfaces was studied through the determination of culturable and total cells using colony-forming units (CFU) enumeration (Figure 2) and flow cytometry (FC) analysis (Appendix A). Both functionalized coatings displayed comparable antimicrobial performance. Culturability analysis showed that both PU-based coatings containing cyclam-based Fe(III) complexes significantly reduced the number of biofilm culturable cells by 41% and 50%, respectively, compared to the control (*p* < 0.05, Figure 2a). A similar trend was observed for total biofilm cells, as *C. marina* biofilms developed on **PU/FeCy-1** and **PU/FeCy-2** surfaces presented, on average, 31% and 38% lower number of total cells, respectively, than **PU** (control) (*p* < 0.05, Figure 2b), demonstrating their antibiofilm performance. Despite an average difference of 8% in the percentages of reduction between the two PU/FeCy surfaces for both variables, this was not significant (*p* > 0.05), suggesting that the N-functionalization of the cyclam backbone with CH_2_CH_2_CH_2_OH pendant arms on **FeCy-2** did not significantly enhance its antimicrobial effect against *C. marina* under the tested experimental conditions.

The tested surfaces were also analyzed concerning their water contact angles (*θ_W_*) and roughness, since bacterial adhesion and subsequent biofilm formation can be influenced by the hydrophobicity of both the bacterium and the surface, and the topography of the surface materials, respectively [23].

Representative water contact angle images of the surfaces are presented in Figure 3. The *θ_W_* values of **PU** (control), **PU/FeCy-1**, and **PU/FeCy-2** surfaces were 91.3 ± 3.2°, 94.3 ± 4.0°, and 98.6 ± 4.8°, respectively, indicating that all surfaces presented a slightly hydrophobic character (*θ_W_* > 90°). Moreover, the *θ_W_* of **PU/FeCy-2** was significantly higher than that of **PU/FeCy-1** (*p* = 0.042), suggesting that the introduction of pendant arms with long aliphatic chains in the macrocycle is responsible for the increase in hydrophobicity. Previous studies reported that *C. marina* is hydrophilic [24,25], which suggests that, in theory, its adhesion to both antimicrobial surfaces is thermodynamically unfavorable, since their *θ_W_* values were higher than that of the control (*p* = 0.084 and *p* = 0.001 for **PU**/**FeCy-1** and **PU/FeCy-2**, respectively). Thus, biofilm formation may have been hindered by the more hydrophobic character of the PU-based surfaces containing the cyclam-based Fe(III) complexes **FeCy-1** and **FeCy-2** compared to the control.

Figure 4 shows representative 2D and 3D Atomic Force Microscopy (AFM) images, as well as two roughness parameters (absolute average (R_a_) and root mean square (R_q_)) for the three surfaces under study. **PU** (control) exhibited a relatively homogeneous and smooth surface, with an average roughness value of 4.05 ± 0.32 nm (R_q_ = 6.79 ± 0.72 nm). The **PU/FeCy-1** samples demonstrated a slight increase in roughness (R_a_ = 5.29 ± 4.22 nm, R_q_ = 7.30 ± 5.47 nm), suggesting the formation of surface structures or the presence of aggregates, likely due to incomplete dispersion of the iron complex within the polymer matrix. These structures appeared to be visible in the sample photographs obtained using the OCT camera (Figure 3b). The **PU/FeCy-2** sample exhibited a reduction in roughness (R_a_ = 3.59 ± 0.21 nm, R_q_ = 4.92 ± 0.73 nm) compared to **PU/FeCy-1**, but some surface irregularities were also detected by AFM (Figure 4) and OCT (Figure 3b). Although numerical variations were observed in the roughness parameters among **PU**, **PU/FeCy-1**, and **PU/FeCy-2**, these differences fall within the margin of error, indicating that the modifications introduced by cyclam-based Fe(III) complexes did not substantially alter the surface topography.

Given the similarity of the *θ_W_* values between the surfaces and the lack of statistically significant changes in roughness, the antimicrobial properties of the immobilized compounds are more likely to explain the differences registered on the antibiofilm effect of **PU/FeCy-1** and **PU/FeCy-2** surfaces in long-term biofilm development rather than surface thermodynamics and topography.

Regardless of the antibiofilm performance of the surfaces, the reduction in the number of culturable cells was higher than that of the total cells, suggesting a killing effect associated with the cyclam-based Fe(III) complexes **FeCy-1** and **FeCy-2** incorporated into the coating formulations. The antibacterial activity of analogous cyclam-derived salts was tested against *S. aureus* and *E. coli*, where the *trans*-disubstituted cyclam salts with trifluoromethylbenzyl substituents on the nitrogen atoms of the macrocyclic ring demonstrated the highest activity [26]. According to the authors, the antimicrobial activity of these cyclam-based salts may be due to electronic or stereochemical factors, which are affected by the chemical nature and the polarity of the benzyl groups [26]. The electronic delocalization created by the polar CF_3_ groups and the aromatic rings of the [{H_2_(^4-CF3^PhCH_2_)_2_Cyclam}FeCl_2_]Cl, **FeCy-1**, and [{(HOCH_2_CH_2_CH_2_)_2_(^4-CF3^PhCH_2_)_2_Cyclam}FeCl_2_]Cl, **FeCy-2**, compounds may thus govern the electronic interactions between the molecule and the receptor, affecting the antibacterial activity [26].

Moreover, to better understand the antibacterial properties of the cyclam-based Fe(III) complexes, their effects on bacterial metabolic activity and reactive oxygen species (ROS) production were evaluated using flow cytometry (FC). *C. marina* cells were grown on **PU**, **PU/FeCy-1,** and **PU/FeCy-2** surfaces for 24 h, and then stained with a metabolic activity marker (5(6)-carboxyfluorescein diacetate (5-CFDA)) and an ROS indicator (2′,7′-dichlorofluorescein diacetate (DCFH-DA)). Biofilm cells exposed to **PU/FeCy-1** and **PU/FeCy-2** surfaces, stained with 5-CFDA, exhibited a 1.8- and 1.6-fold higher mean intensity of fluorescence (MIF), respectively, compared to those grown on the PU control surface. This increase in MIF resulted from a shift in the fluorescence of the cell populations exposed to the PU-modified surfaces (Appendix A), indicating that *C. marina* cells undergo changes in metabolic activity upon exposure to **PU/FeCy-1** and **PU/FeCy-2** surfaces. In parallel, staining of *C. marina* biofilm cells with DCFH-DA (Appendix A) revealed that exposure to PU-modified surfaces induced ROS production, as demonstrated by a 2.8- and 2.3-fold increase in the MIF of cells exposed to **PU/FeCy-1** and **PU/FeCy-2** surfaces, respectively, compared to those grown on the control surface. It is therefore possible that the marine bacteria increased their metabolism as a consequence of the oxidative stress caused by ROS production upon exposure to the cyclam-based Fe(III) coatings. In fact, bacterial stress responses are regulated at different cellular levels, leading to changes in gene expression, protein activity, and cellular metabolism [27].

Mature biofilms are characterized by a diversity of structures and channels that facilitate diffusion and mass transport, and also affect the shear forces on their surface [28], thereby affecting biofilm resistance against mechanical or chemical stresses [29,30,31]. Therefore, a set of qualitative and quantitative parameters (thickness, biovolume, porosity, and contour coefficient) was investigated to characterize the structure of the biofilms using Optical Coherence Tomography (OCT), since these parameters significantly affect underwater device performance [32]. Representative 3D cross-sectional images showing the spatial distribution of biofilm structures across **PU** (control), **PU/FeCy-1**, and **PU/FeCy-2** surfaces after 42 days of incubation are shown in Figure 5. The modified surfaces were clearly more effective in inhibiting biofilm growth during the 6-week assay and displayed visible disparities in their structures compared with those grown on **PU** (control). A qualitative analysis of Figure 5 suggests that biofilms formed on the modified surfaces were more homogeneous, flatter, and thinner than those formed on the **PU** control surface. However, no considerable visual differences were observed between the 3D images of the biofilms formed on **PU/FeCy-1** and **PU/FeCy-2**. To clarify these observations, a full quantitative analysis of the thickness, biovolume, porosity, and contour coefficient was performed, and the results are presented in Figure 6.

The antibiofilm potential of the **PU/FeCy-1** and **PU/FeCy-2** surfaces was demonstrated by a significant decrease in biofilm thickness and biovolume compared to the control (Figure 6a,b). The thickness values obtained were similar (83 ± 20 µm and 77 ± 26 µm for **PU/FeCy-1** and **PU/FeCy-2**, respectively) and significantly lower than those of the control **PU** (199 ± 64 µm, *p* < 0.05, Figure 6a). This represents reductions of 58% and 61% for **PU/FeCy-1** and **PU/FeCy-2**, respectively. Accordingly, the biovolume was significantly lower for biofilms formed on the PU-based coatings containing 1 wt.% of complexes **FeCy-1** and **FeCy-2** (*p* < 0.05, Figure 6b), achieving reductions of 50% for **PU/FeCy-1** and 60% for **PU/FeCy-2.**

The percentage of biofilm porosity was also determined (Figure 6c), as the spatial localization of microorganisms can impact biofilm development by affecting mass transport across pore spaces [33]. Results showed a significant reduction in biofilm porosity for both functionalized surfaces (95% for **PU/FeCy-1** and 97% for **PU/FeCy-2**) in comparison to the control surface **PU** (*p* < 0.05). Taken together with the results obtained for biofilm total and culturable cells (Figure 2), a decrease in porosity may contribute to fewer and smaller empty spaces within the biofilm (such as pores and channels), which may hamper mass transfer (oxygen and nutrients) to the deeper layers of the biofilm, potentially causing higher levels of stress [34], diminishing their viability, and impairing cell growth [24,34,35].

Additionally, the contour coefficient of *C. marina* biofilms was determined (Figure 6d) as a measure of the extent of biofilm exposed to the surrounding medium. Contour coefficient values close to 1 reflect a homogeneous and flat biofilm, and as they vary from 1, biofilm structures become more irregular [36]. The values obtained for biofilms formed on PU-based surfaces containing the FeCy complexes **FeCy-1** and **FeCy-2** were significantly closer to 1 (2.0 ± 0.5 for **PU/FeCy-1** and 1.8 ± 0.7 for **PU/FeCy-2**) than those formed on **PU** (8.4 ± 2.5), which indicates more uniform and flatter biofilms. The contour was approximately 77% lower for the PU/FeCy surfaces than that for the **PU** control (*p* < 0.05). These differences in biofilm homogeneity are visible in Figure 5. The smoother top of the biofilms grown on the coating formulations containing **FeCy-1** and **FeCy-2** complexes resulted in a lower contact area available for the penetration of nutrients and oxygen into the biofilm and adhesion of other organisms [36]. Furthermore, homogeneous biofilm structures are often associated with greater cohesion [36], which can induce limitations in mass transfer towards the bottom of biofilms [37]. Higher shear rate values have also been determined for smoother biofilms, as irregularities sometimes cause stagnant zone formation [28]. Similar effects on the spatial distribution of *C. marina* biofilm structures have been observed in previous long-term studies using other polymeric matrices incorporating different antifouling agents [24,36,38].

In line with the results obtained for cell viability (Figure 2), no significant differences were observed in the performance of PU/FeCy-coated surfaces for any of the biofilm parameters analyzed by OCT. However, the tendency indicated that **PU/FeCy-2** exhibited slightly higher reductions in all parameters than **PU/FeCy-1**. The additional N-functionalization of the cyclam backbone with CH_2_CH_2_CH_2_OH pendant arms may have increased the lipophilic character of the complex, facilitating contact with bacterial cells and subsequent penetration through cell membranes.

To reinforce and corroborate the antibiofilm efficacy of the PU-based coating formulations evidenced by OCT analysis, the differences in *C. marina* biofilm distribution across the three tested surfaces over 42 days were assessed through Confocal Laser Scanning Microscopy (CLSM). Figure 7 displays representative 3D images showing the aerial view of the biofilms and the shadow projection representing the biofilm thickness. Biofilm growth was more evident on the **PU** control surface, which was covered by a thick biofilm, whereas the biofilms developed on the PU/FeCy-based coatings were covered by a significantly lower amount of biofilm. Additionally, the surface area covered by the biofilm on the **PU/FeCy-2** surface was smaller than that on **PU/FeCy-1**.

The results obtained by OCT and CLSM for the biofilm architecture were consistent. All qualitative and quantitative parameters indicated a significant long-lasting antifouling effect on the growth of *C. marina* biofilms.

Additionally, to assess the iron leaching from FeCy-based surfaces, an Inductively Coupled Plasma Optical Emission Spectroscopy (ICP-OES) analysis was performed to the water solutions (leaching waters) that have been in contact with **PU/FeCy-1** and **PU/FeCy-2** surfaces for 3 days, which reflects the average period at which the culture medium was replaced. Iron concentrations of 11.6 × 10^−3^ µg·mL^−1^ and 0.8 × 10^−3^ µg·mL^−1^ were detected on the water in contact with the **PU/FeCy-1** and **PU/FeCy-2** surfaces, respectively. These findings indicate residual metal leaching from the PU-based marine coatings [39], suggesting a potential long-term effectiveness and low environmental impact, in accordance with the permissible limits for iron in drinking water systems established by the WHO [2]. On the other hand, the ecotoxicity of the cyclam-based Fe(III) complexes, expressed as the 50% effective concentration (EC_50_) against *Daphnia magna*, was determined to be 11.6 µg·mL^−1^ for **FeCy-1** and 6.5 µg·mL^−1^ for **FeCy-2**. This highlights that in the event of cyclam complexes leaching, their toxicity is negligible, further supporting the long-term effectiveness and low environmental impact of the PU/FeCy coatings.

## 3. Materials and Methods

### 3.1. Synthesis of Cyclam-Based Fe(III) Complexes

Complexes [{H_2_(^4-CF3^PhCH_2_)_2_Cyclam}FeCl_2_]Cl, **FeCy-1**, and [{(HOCH_2_CH_2_CH_2_)_2_(^4-CF3^PhCH_2_)_2_Cyclam}FeCl_2_]Cl, **FeCy-2**, were prepared according to previously published procedures [16,21,40]. A detailed description for the preparation of both complexes is also presented herein as Appendix A.

### 3.2. Formulation of PU-Based Marine Coatings with Cyclam-Based Fe(III) Complexes

The **FeCy-1** and **FeCy-2** complexes were incorporated into a two-component commercial polyurethane (PU)-based marine paint at concentrations of approximately 1.0 wt.% (Table 1), following previously established methodologies [11]. This was performed to assess the antifouling potential of cyclam-based Fe(III) complexes as antibiofilm agents in surface coatings. The paint system utilized the base resin F0032 and the curing agent 95,580 (Hempel, A/S, Copenhagen, Denmark).

For the preparation of PU-based formulations, complexes **FeCy-1** and **FeCy-2** were first dissolved in N-methyl pyrrolidone (NMP, 99.5%, Acros Organics, Geel, Belgium) at a complex/solvent weight ratio of 0.20. This solution was then added and blended into the PU-based paint components in exact amounts to yield a complex content of approximately 1 wt.% in the wet and uncured coating formulations (c.f. Table 1).

The weight ratio of the base resin to the curing agent in all prepared formulations was 9:1, following the supplier’s recommendations.

The resulting PU/complex-based formulations were used to coat 1 cm^2^ acrylic substrates (Probalplás, Portugal) using a 4-sided film applicator (200 μm) with a reservoir (TQC Sheen), following the procedure recommended by the paint supplier. Substrate coating involved a single coating step and curing at room temperature (23 ± 2 °C) for about one week to ensure complete curing of the polymeric coating matrix.

### 3.3. Water Contact Angle Measurements

The hydrophobicity of the PU-based substrates was estimated by measuring the water contact angle via the sessile drop method employing a contact angle meter (OCA 15 Plus, DataPhysics Instruments, Filderstadt, Germany) as previously described [38]. These measurements were conducted at room temperature (25 ± 2 °C), with a minimum of ten determinations made for each surface. If the water contact angle (*θ_W_*) is lower than 90°, the material is considered hydrophilic. Conversely, if *θ_W_* > 90°, the material is hydrophobic [41].

### 3.4. Atomic Force Microscopy (AFM)

The topographical properties of the **PU**, **PU/FeCy-1,** and **PU/FeCy-2** surfaces were evaluated by AFM using a PicoSPM LE system (Molecular Imaging) and Agilent Technologies PicoView 1.14.4 software (Keysight Technologies, Santa Rosa, CA, USA). The images were obtained in air, at room temperature, in HQ:NSC35/Hard/Al BS-C μmasch^®^ dynamic mode. The surface roughness values were obtained using Gwyddion^®^ software (version 2.67) and represent the average values calculated from measurements taken on three distinct samples.

### 3.5. Inductively Coupled Plasma Optical Emission Spectroscopy (ICP-OES)

ICP-OES was employed to quantify the leaching of iron from the **PU/FeCy-1** and **PU/FeCy-2** surfaces. The surfaces were exposed to 15 mL of ultrapure water for 3 days under biofilm formation conditions (25 °C, 185 rpm), reflecting the average period at which the culture medium was replaced. The resulting solutions were analyzed using an ICP-OES ICAP 7400 THERMO (Waltham, MA, USA), equipped with a nebulizer system and optical emission spectroscopy for detection. Metal concentration in the analyzed samples was quantified using a calibration curve specifically prepared for iron. All measurements were carried out in triplicate.

### 3.6. Ecotoxicity of Cyclam-Based Fe(III) Complexes

The ecotoxicity of **FeCy-1** and **FeCy-2** complexes was evaluated using the *Daphnia magna* acute immobilization test, following the ISO 6341:2012 standard [42]. The tests were conducted in the accredited Laboratório de Análises do Instituto Superior Técnico in Lisbon, Portugal. Ecotoxicity was quantified as the effective concentration (EC_50_), representing the concentration at which 50% of the test organisms exhibited a defined toxic effect.

### 3.7. Bacterial Strain and Culture Preparation

To evaluate the performance of **FeCy-1** and **FeCy-2** complexes as antibiofilm formulations in marine surface coatings, *C. marina* DSM 4741 (Leibniz Institute DSMZ, Braunschweig, Germany) was used as a model marine bacterium due to its ubiquity in coastal seawater and strong biofilm-forming abilities [25,43,44]. *C. marina* aliquots cryo-preserved in Våatanen Nine Salt Solution (VNSS) medium, formulated to replicate marine nutritional conditions [45], were streaked on VNSS agar plates (VNSS medium supplemented with 15 g·L^−1^ agar; VWR International, Leuven, Belgium) and incubated for 24 h at 25 °C. To prepare the starting cultures, single colonies were inoculated in 250 mL of VNSS medium and incubated overnight at 25 °C in an orbital shaker at 120 rpm (25 mm orbital diameter; Agitorb 200ICP, Norconcessus, Ermesinde, Portugal). Subsequently, the overnight cultures were centrifuged (Eppendorf Centrifuge 5810R, Eppendorf, Hamburg, Germany) at 3772× *g* for 10 min, and the resulting pellet was reconstituted in sterile VNSS medium to an optical density at 610 nm of 0.2, yielding a bacterial suspension with a concentration of 1 × 10^8^ CFU·mL^−1^.

### 3.8. Biofilm Formation Assay

The colonization capacity of *C. marina* on PU-based coatings was evaluated after 42 days (6 weeks) through biofilm formation assays using 12-well microplates (VWR International, Carnaxide, Portugal) under controlled hydrodynamic conditions. Prior to each experiment, all surfaces were sterilized for 30 min under ultraviolet radiation. Subsequently, the surfaces were fixed onto the microplate wells using double-sided adhesive, and each well was inoculated with 3 mL of the *C. marina* cell suspension prepared as previously described. The microplates were then incubated at 25 °C in an orbital shaker at 185 rpm. An average shear rate of 40 s^−1^ and a maximum of 120 s^−1^ were attained [37,38], encompassing the shear rate of 50 s^−1^ reported for a ship in a harbor [46]. These shear rate ranges can also be observed in equipment and devices exposed to similar marine environments [37]. Previous evidence has indicated that biofilm development in this system yields results comparable to those observed during extended surface immersion in seawater [47]. Biofilms were examined after 6 weeks of development, aligning with approximately half of the minimum economically viable timeframe deemed necessary for the maintenance and cleaning of underwater monitoring systems [24,48]. During this incubation period, the culture medium was replaced twice per week. The experiments were performed in three independent biological replicates, each comprising three technical replicates (n = 9).

### 3.9. Biofilm Analysis

At the end of each experiment, the culture medium was carefully removed from the wells, and biofilms were gently rinsed with 3 mL of sterile sodium chloride solution (8.5 g·L^−1^ NaCl) to remove loosely attached bacteria. Finally, biofilms were analyzed regarding the number of culturable and total cells by CFU enumeration and FC analysis, respectively, and were imaged by OCT and CLSM.

#### 3.9.1. Enumeration of Culturable and Total Cells

Following the removal of non-adherent cells, the PU-based surfaces were inserted in 15 mL Falcon tubes containing 2 mL of sterile NaCl solution, and biofilm cells were displaced from the coupons by vortexing for 2 min at maximum power (ZX4, Velp Scientifica, Usmate, Italy). The obtained biofilm cell suspensions were then serially diluted in NaCl solution.

The biofilm culturable cells were enumerated by CFU counting (CFU·cm^−2^) by plating the diluted bacterial suspensions on VNSS medium and incubating for 24 h at 25 °C. Total cells (cells·cm^−2^) were quantified through FC (CytoFLEX V0-B3-R1, Beckman Coulter, Brea, CA, USA), where 10 µL of the diluted biofilm suspensions was acquired at a flow rate of 30 µL·min^−1^. Data were retrieved using the CytExpert software (version 2.4.0.28, Beckman Coulter, Brea, CA, USA).

#### 3.9.2. Visualization of Spatial Organization and Quantitative Structural Parameters of Biofilms

Qualitative and quantitative structural characterization of biofilms was determined through OCT analysis (Thorlabs Ganymede Spectral Domain Optical Coherence Tomography system with a central wavelength of 930 nm, Thorlabs GmbH, Dachau, Germany), as reported by Romeu et al. [36,37]. Prior to OCT analysis, the wells were filled with 3 mL of sterile NaCl solution to ensure biofilm hydration. Afterwards, three-dimensional (3D) imaging of *C. marina* biofilms developed on PU-based surfaces was performed as previously described [37]. Given that biofilms primarily consist of water [49], the refractive index was set to 1.40, resembling that of water (1.33). Each coupon was imaged in four different fields of view to guarantee the accuracy and consistency of the results. The 3D images of biofilms were processed and analyzed using a routine developed in the Image Processing Toolbox from MATLAB 8.0 and Statistics Toolbox 8.1 (The MathWorks, Inc., Natick, MA, USA) [36,37]. The quantitative structural parameters of the biofilms, namely, thickness, porosity, contour coefficient, and biovolume [36], were calculated as reported by Romeu et al. [36].

The spatial organization of biofilms was also assessed through CLSM (Leica TCS SP5 II confocal laser scanning microscope, Leica Microsystems, Wetzlar, Germany). At the end of the experiment, biofilms were stained in green with SYTO 9 (Thermo Fisher Scientific, Waltham, MA, USA) and scanned at an excitation wavelength of 488 nm (argon laser) using a 40× water objective (Leica HCX PL APO CS, Leica Microsystems, Germany). For each sample, stacks of horizontal plane images (512 × 512 pixels, equivalent to 387.5 × 387.5 µm) were captured with a z-step of 1 µm from four randomly selected areas. For image analysis, the “Easy 3D” function of IMARIS 9.3.1 software (Bitplane, Zürich, Switzerland) was used to reconstruct the 3D projections of biofilms.

### 3.10. Effects of PU-Modified Surfaces on Bacterial Metabolic Activity and ROS Production

The effects of the **PU/FeCy-1** and **PU/FeCy-2** surfaces on *C. marina* metabolic activity and ROS production were characterized by flow cytometry (FC). Bacteria were exposed to the synthesized surfaces at 25 °C and 185 rpm for 24 h. Subsequently, biofilm cell suspensions, obtained as described in Section 3.9.1, were stained in the absence of light for 30 min with 5-CFDA (Sigma-Aldrich, Taufkirchen, Germany) at 5 µg·mL^−1^ to measure cell metabolic activity, and DCFH-DA (Sigma-Aldrich, Taufkirchen, Germany) at 10 µM to detect endogenous ROS production, as previously described [24]. 5-CFDA is a nonfluorescent, lipophilic substrate that is hydrolyzed by esterases in the cytoplasm of metabolically active cells, forming the fluorescent compound carboxyfluorescein [50]. Similarly, DCFH-DA is a nonfluorescent, cell membrane-permeable molecule that is hydrolyzed by intracellular esterases to produce dichlorodihydrofluorescein (DCFH), which is then oxidized by ROS, producing the fluorescent compound 2′,7′-dichlorofluorescein (DCF) [51].

Biofilm cells were analyzed by acquiring 20,000 cells at a flow rate of 30 µL·min^−1^ in the CytoFLEX flow cytometer model V0-B3-R1. Data were analyzed using CytExpert software.

### 3.11. Statistical Analysis

Statistical analysis was performed using the IBM SPSS Statistics version 26 (IBM SPSS, Inc., Chicago, IL, USA). Descriptive statistics were used to calculate the mean and standard deviation from at least three independent experiments with technical triplicates for the number of culturable and total cells, and biofilm thickness, porosity, contour coefficient, and biovolume. Since the variables were not normally distributed, a nonparametric analysis using the Mann–Whitney test was performed to assess the differences between the results obtained for **PU** (control) and the FeCy-modified coating surfaces (**PU/FeCy-1** and **PU/FeCy-2**) for each of the variables. Statistically significant differences were considered for *p*-values < 0.05, corresponding to a confidence level of 95%.

## 4. Conclusions

This study demonstrated that novel PU-based coated surfaces containing cyclam-based Fe(III) complexes with formula [{R_2_(^4-CF3^PhCH_2_)_2_Cyclam}FeCl_2_]Cl (R = H, HOCH_2_CH_2_CH_2_) were significantly less prone to *C. marina* biofilm development than the control surface without the complexes. They not only decreased the culturable and total cell numbers, but also reduced the amount of *C. marina* biofilms formed after 42 days. Additionally, exposure to cyclam-based Fe(III) coatings increased bacterial metabolism as a result of oxidative stress caused by ROS production.

These findings demonstrated the strong potential of cyclam-based Fe(III) complexes as bioactive additives in commercial PU-based coating formulations, effectively preventing bacterial colonization in marine environments. This is particularly important considering the growing demand for long-lasting antifouling solutions that can extend the service life of marine materials. With the ability to inhibit biofilm formation, these coatings offer a promising strategy to mitigate marine biofouling. This capability is vital for the sustainability of marine operations, as extending material lifespan significantly reduces the environmental and economic burden of frequent repairs and replacements.

## Figures and Tables

**Figure 1 molecules-30-00917-f001:**
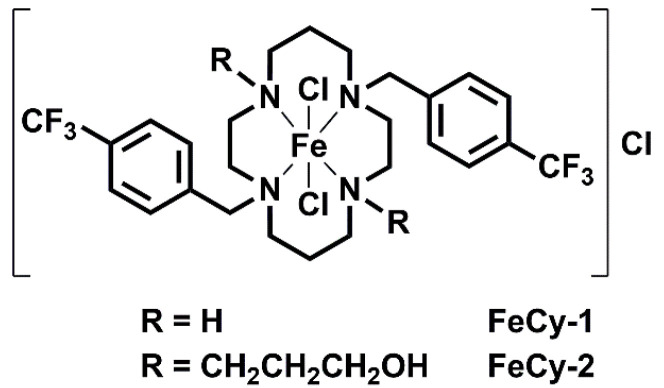
Chemical structure of [{H_2_(^4-CF3^PhCH_2_)_2_Cyclam}FeCl_2_]Cl (**FeCy-1**) and [{(HOCH_2_CH_2_CH_2_)_2_(^4-CF3^PhCH_2_)_2_Cyclam}FeCl_2_]Cl (**FeCy-2**).

**Figure 2 molecules-30-00917-f002:**
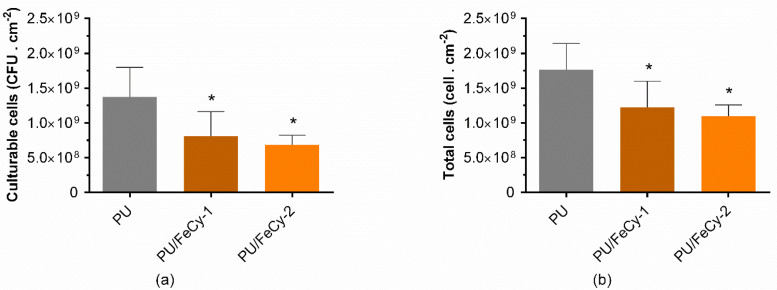
(**a**) Culturable and (**b**) total cells of *C. marina* biofilms formed on **PU** (control), **PU/FeCy-1**, and **PU/FeCy-2** surfaces after 42 days. The asterisks represent statistical differences between PU and the PU/FeCy surfaces (*p*-values < 0.05).

**Figure 3 molecules-30-00917-f003:**
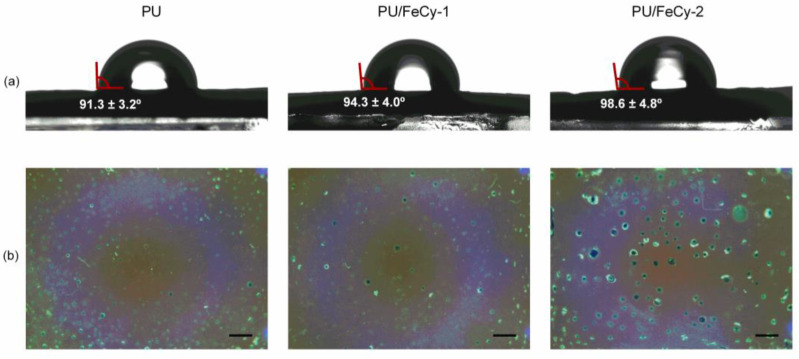
(**a**) Representative images of water contact angle measurements and (**b**) visual depictions (captured by the Optical Coherence Tomography (OCT) camera; scale bar = 1 mm) of **PU** (control), **PU/FeCy-1**, and **PU/FeCy-2** surfaces.

**Figure 4 molecules-30-00917-f004:**
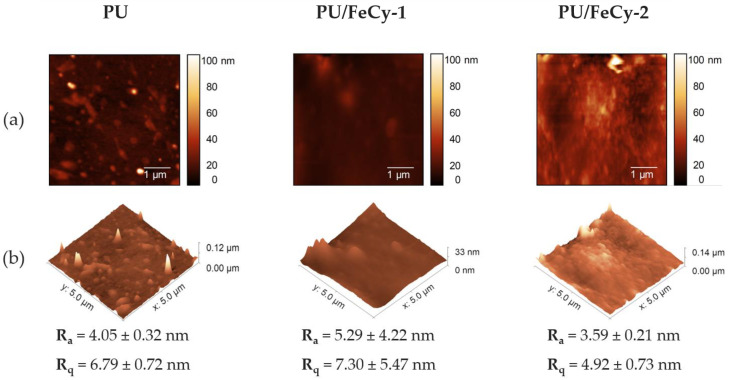
(**a**) Two-dimensional and (**b**) three-dimensional AFM images of **PU** (control), **PU/FeCy-1**, and **PU/FeCy-2** surfaces, including absolute average (R_a_) and root mean square (R_q_) values. All images correspond to a 5 × 5 µm^2^ surface area.

**Figure 5 molecules-30-00917-f005:**
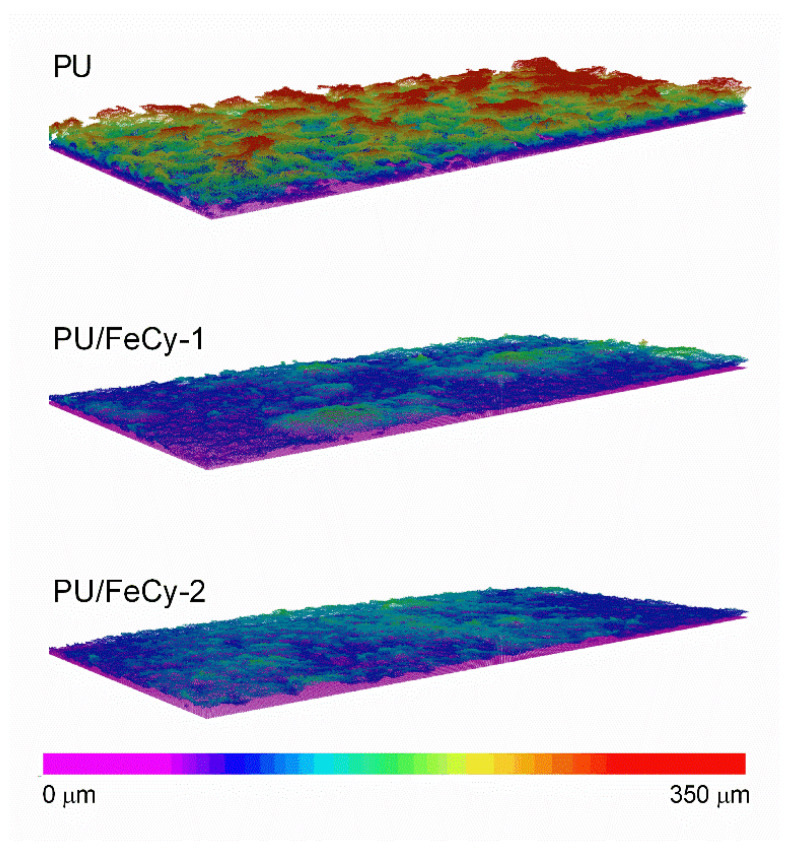
Representative 3D OCT images of C. marina biofilms formed on **PU** (control), **PU/FeCy-1**, and **PU/FeCy-2** surfaces after 42 days. The color scale shows the range of biofilm thickness. All images were obtained in a scan range of 2490 µm × 1512 µm × 600 µm.

**Figure 6 molecules-30-00917-f006:**
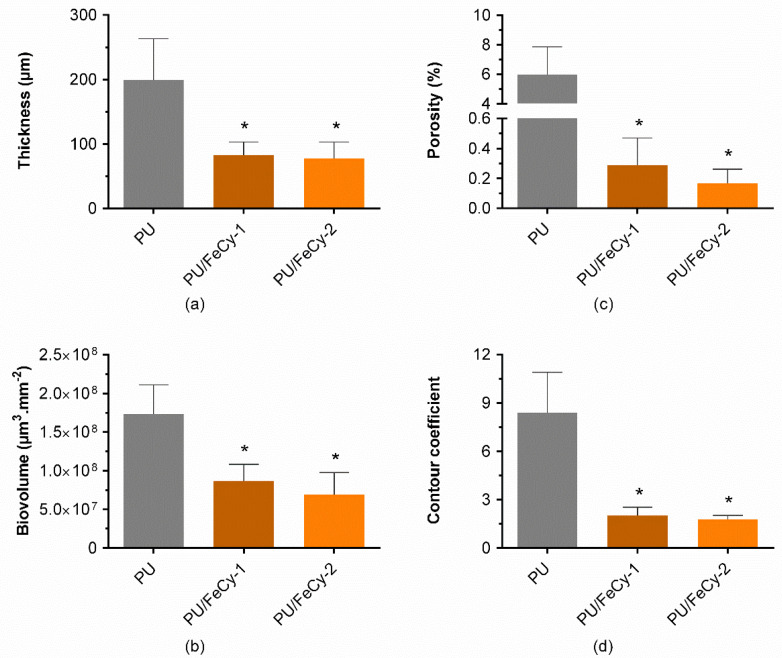
(**a**) Thickness, (**b**) porosity, (**c**) contour coefficient, and (**d**) biovolume of *C. marina* biofilms formed on **PU** (control), **PU/FeCy-1**, and **PU/FeCy-2** surfaces after 42 days. The asterisks represent statistical differences between **PU** and PU/FeCy surfaces (*p*-values < 0.05).

**Figure 7 molecules-30-00917-f007:**
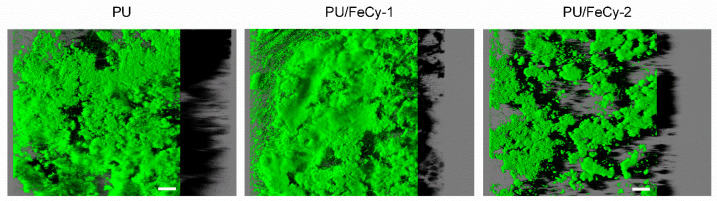
CLSM images of C. marina biofilms on **PU** (control), **PU/FeCy-1**, and **PU/FeCy-2** surfaces after 42 days. These representative images were obtained from confocal z-stacks using the IMARIS 9.3.1 software and present an aerial, 3D view of the biofilms, with the shadow projection on the right. The white scale bars represent 40 μm.

**Table 1 molecules-30-00917-t001:** Marine coating formulations containing cyclam-based Fe(III) complexes.

Coating Formulation	Base/Curing Agent Ratio (*v*/*v*)	Complex Content (wt.%)	Complex/Solvent Ratio (m/m)
**PU** (Control)	9.0 ± 0.1	----	----
**PU/FeCy-1**	8.9 ± 0.1	1.03 ± 0.02	0.20 ± 0.01
**PU/FeCy-2**	8.9 ± 0.1	0.99 ± 0.02	0.19 ± 0.01

**PU** = Polyurethane-based marine paint; **FeCy-1** = [{H_2_(^4-CF3^PhCH_2_)_2_Cyclam}FeCl_2_]Cl; **FeCy-2 =** [{(HOCH_2_CH_2_CH_2_)_2_(^4-CF3^PhCH_2_)_2_Cyclam}FeCl_2_]Cl; N-methyl pyrrolidone as solvent.

## Data Availability

The original contributions presented in this study are included in the article. Further inquiries can be directed to the corresponding authors.

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
