# Peer review of "New Cyclam-Based Fe(III) Complexes Coatings Targeting Cobetia marina Biofilms"

_molecules, 2025, doi:10.3390/molecules30040917_

Round 1
Reviewer 1 Report
Comments and Suggestions for Authors
The manuscript investigated the antifouling performance of novel polyurethane (PU)-based coatings containing cyclam-based Fe(III) complexes against Cobetia marina biofilm formation over 42 days under mimicked marine environments which is quite interesting. However, the manuscript appears to be just a part of a study rather than a complete project. My specific questions are as follows:
1. Specific information about the reagents should be provided in the text.
2. Abbreviations should appear at the position where the corresponding words first appear in the text.
3. Many parts in the text are not described clearly or may cause confusion for readers. For example, it is necessary to list specific methods in lines 168 - 170 and lines 57 - 60.
4. Please provide representative contact angle pictures, and it is recommended to provide actual photos of the samples.
5. How was the molecular structure of the mixture in Figure 1 determined? What was the specific reduced pressure? Please explain in detail.
6. The reviewer doubts the significance analysis of the author, especially in the part of Figure 2.
7. Most of the manuscript only describes the experimental data, and the explanations for the causes are insufficient. It is very necessary to strengthen the discussion part of the paper.
8. The specific process of modifying the substrate with PU/complex-based marine paint as well as physical and chemical properties characterization of modified surfaces are lacking. Only a brief introduction was made in lines 280 - 284, which makes it difficult to determine whether the subsequent antibiofilm experimental results are derived from the microstructural parameters/surface substances on the surface.
Author Response
Please see our responses in the attached document.

Reviewer 2 Report
Comments and Suggestions for Authors
Following are some questions and suggestions that need to be discussed in the manuscript by the author;
1. What is the minimum inhibitory concentration (MIC) of FeCy-1 and FeCy-2 against the tested pathogens? The authors need to determine the MIC values before selecting appropriate sub-MIC concentrations to evaluate the antibiofilm efficacy of these compounds before coating.
2. The study reports significant reductions in both culturable and total biofilm cells on the functionalized surfaces, suggesting a potential bactericidal effect. Could the reduction in total cell numbers be attributed to biofilm disruption or structural damage, rather than direct bactericidal activity? Additionally, how was biofilm integrity assessed beyond just measuring cell viability?
3. While biofilm formation on the PU/FeCy surfaces is significantly inhibited, the total cell viability, including non-culturable cells, is only slightly reduced. What could explain this discrepancy, and how might these findings inform the development of more effective antifouling coatings in the future?
4. The paper attributes the antimicrobial effects of the Fe(III) complexes to electronic or stereochemical factors, particularly the influence of CF3 groups. However, could the leakage of metal ions from the complexes also play a role in biofilm inhibition? How can this potential contribution be experimentally tested and distinguished from the proposed electronic effects?
5. OCT and CLSM analyses reveal differences in biofilm thickness, porosity, and homogeneity on the functionalized surfaces. Could these structural changes be directly linked to the mechanism of biofilm inhibition, for example, by inducing a more cohesive biofilm structure that is less susceptible to environmental stresses? How does the structural integrity of the biofilm correlate with its viability?
6. Were there any observable morphological changes in C. marina cells treated with FeCy-1 and FeCy-2? This information could provide further insights into the mode of action of these compounds on microbial cells.
7. To confirm the structure of the synthesized FeCy-1 and FeCy-2 complexes, could the authors provide NMR data for these compounds? Additionally, a detailed synthetic scheme illustrating the preparation of these molecules would help clarify the methodology.
8. What is the stability of the PU-based marine coatings incorporating cyclam-based Fe(III) complexes? It would be valuable to examine the release profile of FeCy-1 and FeCy-2 from the coatings over time to assess their long-term effectiveness and potential environmental impact.
9. How does the presence of FeCy-1 and FeCy-2 affect the growth of C. marina at different time intervals? Understanding the growth dynamics before applying these molecules as coating agents is crucial for evaluating their effectiveness and potential impact on marine ecosystems over extended periods.
Author Response

(The authors gave the same response as above.)

Round 2
Reviewer 1 Report
Comments and Suggestions for Authors
It can be accepted base on this version.
Reviewer 2 Report
Comments and Suggestions for Authors
The author addressed most of the critical questions. Congratulations!